# Two Samples Are Enough: Verbal Confidence Meets Self-Consistency in Reasoning LLMs

## Abstract

Large reasoning models (LRMs) achieve strong problem-solving ability but also produce confident errors, making reliable uncertainty estimation essential. Prior work on standard large language models proposed two approaches: advanced *verbalized confidence* (VC), where the model self-checks its chain-of-thought or directly reports its own certainty, and *self-consistency* (SC), where the agreement across multiple stochastic samples of the answers to the same question indicates reliability. How these methods behave in LRMs, with long, rich, and internally branching reasoning traces, remains unclear.

We present the first systematic evaluation of six VC methods, SC, and their hybrid (VCSC) across nine scientific benchmarks and three LRMs. We find that advanced VC instructions bring little benefit, sometimes reducing accuracy on mathematical tasks, and improving AUROC by about three percentage points on non-mathematical tasks. By contrast, VC-based parallel sampling and hybridization deliver dramatic gains: with just two repeats, VCSC improves AUROC by over 10 points on average. With larger budgets, parallel VC alone can approach perfect discrimination, as in the distilled DeepSeek model on AIME where AUROC reaches 1.0.

These results establish VCSC as a simple, overlooked, and highly effective recipe for uncertainty estimation in LRMs, and deepen our understanding of how these models expose and exploit their own uncertainty.

## 1 Introduction

Large reasoning models (LRMs; e.g DeepSeek-AI et al. 2025, etc.) are rapidly becoming central to scientific and professional applications. Their long chains of thought enable strong problem-solving ability, but errors are often produced with persuasive but misleading certainty. Deploying LRMs safely, therefore, requires not only accurate answers but also reliable confidence estimates.

Prior work on uncertainty estimation (Section 2) has largely focused on non-reasoning large language models (LLMs) and introduced two main approaches: *verbalized confidence* (VC; Tian et al. 2023; Xiong et al. 2023), where models report their own certainty, and *self-consistency* (SC; Wang et al. 2022), where the agreement across multiple stochastic samples of the answers to the same question indicates reliability. Both approaches rest on assumptions about what the reasoning traces reveal. VC assumes the traces surface epistemic cues that support self-verification, while SC assumes that individual traces are narrow, so multiple samples provide a broader coverage. LRMs, however, change the picture: their traces are long, branching, more informative, and often contain explicit uncertainty markers. These shifts suggest that advanced VC methods (Miao et al., 2023; Liu et al., 2025) may benefit from these richer signals, SC may lose some of its relative advantage, and hybrid approaches could capture complementary signals. Yet, beyond vanilla VC, no systematic study has examined how these methods behave in LRMs.

We address this gap with the first comprehensive evaluation of six advanced VC methods, SC, and their combination across nine scientific benchmarks and three LRMs, resulting in the following three contributions:

- **Systematic evaluation.** We conduct the first comprehensive study of six VC methods, SC, and their combination (VCSC) across nine scientific tasks and three LRMs, motivated by the fact that their underlying core assumptions change in the reasoning models.

- **Strong gains from simple hybrids.** We show that combining VC and SC improves the mean AUROC by more than 10 percentage points with only two repetitions. With larger budgets on specific model-task pairs (e.g. distilled DeepSeek model on AIME tasks), the hybrid achieves near-perfect or perfect discrimination.

- **Practical recipe.** Our insights yield actionable guidance: default to vanilla confidence elicitation on mathematical tasks, epistemic elicitation on non-mathematical tasks, but most importantly, avoid relying on a single reasoning trace. Two repeats with VCSC already deliver dramatic and inexpensive gains.

Together, these contributions advance both the understanding of uncertainty estimation abilities of LRMs and our ability to deploy these models safer, by reducing overconfident errors, and more efficiently, by guiding downstream tasks with selective verification or prioritization.

## 2 RELATED WORK

**Self-Consistency and Hybrid Methods.** Self-consistency (SC) estimates reliability from agreement across multiple sampled outputs (Wang et al., 2022). It has seen some attention in standard LLMs and was compared against VC, but always outside the extended reasoning regime (Xiong et al., 2024). Early attempts to combine SC with VC in standard LLMs reported only modest or mixed benefits (Xiong et al., 2024; Huang et al., 2024; Rivera et al., 2024). CoCoA (Vashurin et al., 2025) demonstrated clearer improvements by combining SC with log-probability signals, but this approach requires white-box access and remains limited to standard LLMs, whereas using VC as the confidence signal to supplement SC can be applied in black-box settings. Several systematic reviews likewise do not conclude that VC–SC hybrids provide consistent benefit (Abbasli et al., 2025; Geng et al., 2024; Shorinwa et al., 2025). More recently, Podolak & Verma (2025) argue that confidence and sampling may not be complementary in reasoning models, suggesting that long-form chains of thought already cover much of the solution space SC is meant to expose. Thus, our study is the first to directly study SC and VCSC hybrid in LRMs and overturn these prior intuitions.

**Verbalized Confidence.** Early work studied *vanilla VC*, where models directly report their own certainty (Kadavath et al., 2022; Tian et al., 2023). More elaborate prompting strategies include Fact-and-Reflection (Zhao et al., 2024) and "global SelfCheck" variants (Miao et al., 2023; Manakul et al., 2023; Madaan et al., 2023), which encourage verification or iterative self-feedback but showed mixed benefits. We revisit these ideas for LRMs with our Verification Judge (VeJu). Other work has examined epistemic markers as implicit cues (Liu et al., 2025), which we adapt to the reasoning setting with our Epistemic Judge (EpJu). Most recently, vanilla VC has been evaluated in LRMs, where longer reasoning traces appear to surface more faithful confidence cues than in standard LLMs (Yoon et al., 2025; Zeng et al., 2025). In this work we confirm that reasoning LLMs are generally competitive in VC and further extend this insight to SC. Our primary contributions are also complementary: we ask whether vanilla VC in LRMs can be *further strengthened* with motivated VC variants and VCSC hybrid.

## 3 METHODS

We study confidence estimation methods originally proposed for standard LLMs, adapting them to LRMs and introducing new variants.

### 3.1 VERBALIZED CONFIDENCE METHODS

We consider two families of methods: Elicitation (Xiong et al., 2024), where the model that solves the task also reports its confidence, and Judge (Gu et al., 2025), where a subsequent iteration reads the complete chain and produces a confidence score. Both quantify uncertainty through verbalized confidence (VC), but differ in how the signal is extracted. For this, we consider three uncertainty instruction variants, each mapping responses to a 1–100 confidence grading scale (Table 1). Together,

| Variant | Prompt and grading scale |
|---|---|
| Vanilla | **Instructions:** "Give a confidence number (1–100) representing your overall confidence that the final answer is correct." 
 **Grading scale:** – |
| Verification | **Instructions:** "Give a confidence number (1–100) representing how likely the final answer is correct, based on the quality of the reasoning (soundness, validity, coherence)." 
 **Grading scale:** "1 – fatally flawed; 25 – major gaps; 50 – plausible but unproven; 75 – strong reasoning, probably right; 100 – airtight reasoning, correct." |
| Epistemic | **Instructions:** "Give a confidence number (1–100) representing how confident you are in your final answer (do not re-solve)." 
 **Grading scale:** "1 – random guess; 25 – significant doubts; 50 – mixed feelings; 75 – mostly confident; 100 – completely certain." |

Table 1: Uncertainty instruction prompts used in our main experiments. Each variant is applied in both elicitation and judge methods, with epistemic instruction slightly adapted for the judge (see Appendix B for full prompts).

it results in two vanilla methods and 4 advanced variants. Advanced variants showed limited success in short-trace LLMs. However, more detailed chains of thought in LRMs potentially make their assumptions more plausible, motivating our evaluation of whether VC methods perform substantially better on LRM traces.

***Vanilla elicitation* (VaEl).** Ask the model to provide an answer and a confidence score. This assumes the model can introspectively assess its certainty. Prior work found that this works reasonably well in LLMs (Xiong et al., 2024), and recent studies have shown it transfers strongly to LRMs (Yoon et al., 2025; Zeng et al., 2025).

***Verification elicitation* (VeEl).** Prompt the model to check the validity of its reasoning before assigning confidence. Short LLM traces often lacked sufficient structure for this, whereas LRMs' detailed scratchpads could make self-checks more feasible (Miao et al., 2023).

***Epistemic elicitation* (EpEl).** Steer the model to monitor its certainty by explicitly prompting it to reflect as it reasons. This assumes that extended reasoning budgets allow the model to accommodate both problem-solving and careful self-assessment. Such steering yielded little benefit in LLMs (Tian et al., 2023), but the longer traces of LRMs make the assumption more plausible, motivating an evaluation on this model type.

***Vanilla judge* (VaJu).** Run a second pass that simply reads the reasoning trace and outputs a confidence score. In LLMs, the signal was sparse due to short traces, whereas LRMs offer richer traces that give the judge more evidence to work with (Xiong et al., 2024).

***Verification judge* (VeJu).** Ask the judging iteration to assess whether the reasoning steps are valid and consistent. This relies on traces exposing an explicit logical structure, which LRMs' step-by-step reasoning makes far more realistic than the short outputs of LLMs (Miao et al., 2023).

***Epistemic-markers judge* (EpJu).** Ask the LLM judging pass to attend to hedges, certainty language, and similar cues. Such markers were rare and unreliable in LLMs (Liu et al., 2025), but appear far more frequently in LRMs' extended traces (Venhoff et al., 2025).

Although Epistemic Elicitation and Epistemic-markers Judge use nearly identical instructions, they are expected to behave differently. In Elicitation, the model is prompted to deal with uncertainty markers while solving the problem, which may influence the reasoning process itself. In the Judge-based approach, the model evaluates an already generated chain-of-thought, basing its assessment on explicit uncertainty cues within that reasoning.

Appendix B provides the prompts used for obtaining answers and eliciting confidence.

## 3.2 PARALLEL SAMPLING AND HYBRID METHODS

These methods aggregate multiple samples rather than relying on a single trace. Their core assumption is that a single trace covers only a narrow slice of the solution space, and this is well supported for standard LLMs. In LRMs, however, individual traces already branch widely (Lee et al., 2025; Wan et al., 2025), so the added value of sampling is unclear. Whether SC-based methods remain effective in this setting is therefore an open empirical question, motivating our study. Hybrid methods

combine verbalized confidence and self-consistency signals. While LRMs may weaken SC assumptions and strengthen VC assumptions, it remains unclear whether the two sources of information remain complementary. We introduce a simple weighted combination of SC and VC (VCSC) to test this directly.

***Self-Consistency* (SC).** SC samples $M$ reasoning traces $a_1, \ldots, a_M$ and estimates model's confidence as the fraction of traces that agree with the majority answer $\hat{a}$:

$$\text{SC} = \frac{1}{M} \sum_{i=1}^{M} \mathbf{1}[a_i = \hat{a}].$$

This assumes that each trace explores only a narrow slice of the solution space, so diversity across traces reveals uncertainty. While effective for standard LLMs with short traces, in LRMs each trace already branches internally, weakening the assumption and motivating us to re-examine SC.

***Average Verbalized Confidence* ($\text{VC}_{\text{avg}}$).** Each sampled trace provides an answer $a_i$ with verbalized confidence $c_i$. $\text{VC}_{\text{avg}}$ averages the confidence scores of the traces yielding the majority answer $\hat{a}$:

$$\text{VC}_{\text{avg}} = \frac{1}{|\mathcal{I}_{\hat{a}}|} \sum_{i \in \mathcal{I}_{\hat{a}}} c_i,$$

where $\mathcal{I}_{\hat{a}} = \{i : a_i = \hat{a}\}$. This sharpens confidence estimates compared to a single repeat by pooling across the consistent outputs.

***Verbalized Confidence with Self-Consistency* (VCSC).** SC and $\text{VC}_{\text{avg}}$ may capture complementary signals: SC reflects behavioral agreement, while $\text{VC}_{\text{avg}}$ reflects expressed certainty. VCSC combines them as a weighted average:

$$\text{VCSC} = \lambda \cdot \text{SC} + (1 - \lambda) \cdot \text{VC}_{\text{avg}}$$

In our experiments, we set $\lambda = 0.5$ by default and examine the effect of varying $\lambda$ in Section 5.3.

## 4 EXPERIMENTAL SETUP

**Tasks.** We evaluate the proposed methods on mathematical and non-mathematical reasoning tasks. Mathematical tasks are the primary in-domain setting for reinforcement-learning-based post-training of reasoning LLMs (Ma et al., 2025), so chain-of-thought reasoning in this class of tasks may differ qualitatively from other reasoning domains.

As math tasks, we use the `AIME 2024` and `AIME 2025`[1] datasets (30 problems each) from the American Invitational Mathematics Examination, a widely used benchmark for frontier models. To provide complementary coverage, we also include `GSM8K` (Cobbe et al., 2021), a set of 8.5K diverse grade-school word problems that, while easier than AIME, still remain unsaturated for our models.

As non-mathematical tasks, we use `GPQA Diamond` (Rein et al., 2024), a set of 198 expert-written multiple-choice questions in biology, physics, and chemistry, designed to challenge even domain experts. We also draw from `MMLU-Pro` (Wang et al., 2024), an extension of MMLU (Hendrycks et al., 2021) with harder reasoning questions and larger answer spaces. We focus on the Psychology (798 questions), Health (818), Biology (717), Business (789), and Law (1101) domains, where reliable uncertainty estimates are especially critical.

Together, these datasets cover both structured mathematical reasoning and complex natural and social science reasoning/knowledge domains, providing a comprehensive testbed for uncertainty estimation in LLMs and LRMs.

**Models.** We evaluate three open-source reasoning models: gpt20b-high[2], qwen30b[3], and deepseek8b[4]. The first two are mixture-of-experts models trained with Reinforcement Learning

---

[1] https://artofproblemsolving.com/wiki/index.php/AIME_Problems_and_Solutions
[2] https://huggingface.co/openai/gpt-oss-20b
[3] https://huggingface.co/Qwen/Qwen3-30B-A3B
[4] https://huggingface.co/deepseek-ai/DeepSeek-R1-0528-Qwen3-8B

with Verifiable Rewards (RLVR), while deepseek8b is a fine-tuned dense model that extends the 8b Qwen dense model with supervised finetuning on reasoning traces from DeepSeek-R1 (DeepSeek-AI et al., 2025). The "-high" suffix in gpt20b-high indicates a higher reasoning effort, aligning more closely with the theoretical assumptions of the VC methods we study. These models represent a balanced trade-off: they approach the performance of larger frontier models such as DeepSeek-R1 while remaining light enough to run for up to 100 epochs, ensuring stable and reproducible results. All three support at least 131K tokens of context, enabling long chain-of-thought reasoning and allowing the same model to serve as both task solver and judge in VC-based methods.

**Generation configuration.** We allow models to generate up to 60K tokens for the task solver and an additional 60K tokens for the judge when applicable, ensuring all cases fit within the 131K context window. We use the vLLM framework[5] for all evaluations, adopting its default settings except for temperature and top-$k$. These follow the model authors' recommendations: temperature = 1.0, top-$k$ = 1.0 for gpt20b-high, and temperature = 0.6, top-$k$ = 0.95 for qwen30b and deepseek8b.

**Bootstrap Evaluation Protocol.** For our comparative study of six VC methods (Section 5.1), we generate up to 20 (answer, confidence) tuples per question by running 20 repetitions. Repetitions that fail to follow the answer format are discarded, and questions with no valid tuples are removed: counting unsuccessful extractions as incorrect answers with confidence 0 (mirroring deployment) would unfairly bias confidence metrics toward methods with higher extraction failure rates. We therefore treat these cases as temporary model limitations and exclude them. In practice, this removes one AIME example, 5% of GPQA examples, and up to 1% of examples in other datasets if run for 20 repeats as in Section 5.1, and preserves all examples when we run for 100 repeats and a smaller selection of methods in Section 5.2.

To maximize the utility of available generations, we apply hierarchical bootstrapping: for each bootstrap iteration, we construct a pseudo-dataset by randomly selecting one of the available (answer, confidence) tuples for each question. Metrics are then computed on this pseudo-dataset, and the process is repeated 1000 times. The reported results are the average metrics across these simulated datasets. For self-consistency and hybrid experiments (Section 5.2), we generate up to 100 (answer, confidence) tuples per question. From these, $K$ tuples are sampled per question to form a dataset. Repeating this 1000 times yields 1000 simulated datasets, and we report the mean metric values across them.

This protocol is essential because results for these datasets and models are highly volatile: temperature sampling introduces large variance, which obscures method effects. By combining 20–100 repeats with bootstrapping, we stabilize results toward convergent values, enabling reliable comparisons between methods and ensuring observed differences reflect method performance rather than sampling noise.

**Choice of Confidence Evaluation Metric.** Common metrics for evaluating uncertainty include the Expected Calibration Error (ECE; Guo et al., 2017) and the Area Under the ROC Curve (AUROC; Hanley & McNeil, 1982). AUROC measures how well confidence separates correct from incorrect answers, while ECE measures how closely predicted confidence matches empirical accuracy. Our focus is on the *discriminative power* of confidence signals, as this reflects their informativeness and suitability for downstream calibration or decision-making. Calibration metrics such as ECE, Negative Log-Likelihood (NLL), and Brier score (Brier, 1950) are more appropriate after explicit calibration or in deployment contexts. Using them here would penalize methods that are discriminative but operate on shifted scales (e.g., assigning values only in range 70–100% instead of 0–100%), which is common in LLMs. We therefore use AUROC as our primary evaluation metric, while reporting ECE and NLL in the Appendix A for completeness.

## 5 EXPERIMENTS

### 5.1 COMPARING VERBALIZED CONFIDENCE METHODS

We compare six verbalized-confidence (VC) variants grouped as *elicitation* and *judge* (Section 3.1), evaluating them on mathematical and non-mathematical tasks using accuracy and AUROC. Results

---

[5]https://github.com/vllm-project/vllm

| | AUROC | | | | | | Accuracy | | | | | |
|---|---|---|---|---|---|---|---|---|---|---|---|---|
| Task | VaEl | VeEl | EpEl | VaJu | VeJu | EpJu | VaEl | VeEl | EpEl | VaJu | VeJu | EpJu |
| *gpt20b-high* | | | | | | | | | | | | |
| AIME 2024 | 0.61 | 0.59 | 0.67 | 0.66 | **0.72** | 0.65 | **0.95** | **0.95** | **0.95** | 0.93 | 0.93 | 0.92 |
| AIME 2025 | **0.77** | 0.67 | 0.66 | 0.57 | 0.71 | 0.64 | **0.96** | **0.96** | **0.96** | **0.96** | **0.96** | 0.94 |
| GSM8K | 0.69 | 0.73 | 0.70 | **0.74** | **0.74** | 0.66 | 0.94 | 0.94 | 0.94 | **0.95** | **0.95** | **0.95** |
| *Avg* | *0.69* | *0.66* | *0.68* | *0.66* | *0.72* | *0.65* | *0.95* | *0.95* | *0.95* | *0.95* | *0.95* | *0.94* |
| *deepseek8b* | | | | | | | | | | | | |
| AIME 2024 | **0.88** | **0.88** | 0.86 | 0.83 | 0.86 | 0.84 | 0.77 | 0.74 | 0.73 | **0.79** | **0.79** | **0.79** |
| AIME 2025 | **0.86** | 0.85 | 0.83 | 0.82 | 0.83 | 0.78 | 0.71 | 0.64 | 0.65 | 0.74 | **0.76** | **0.76** |
| GSM8K | 0.71 | 0.70 | 0.70 | **0.73** | **0.73** | 0.67 | 0.93 | 0.93 | 0.93 | 0.93 | 0.93 | 0.93 |
| *Avg* | *0.82* | *0.81* | *0.80* | *0.79* | *0.81* | *0.76* | *0.80* | *0.77* | *0.77* | *0.82* | *0.83* | *0.83* |
| *qwen30b* | | | | | | | | | | | | |
| AIME 2024 | 0.61 | 0.63 | 0.59 | **0.69** | 0.56 | 0.58 | 0.90 | 0.89 | 0.88 | **0.93** | 0.92 | 0.92 |
| AIME 2025 | 0.57 | 0.61 | 0.54 | **0.64** | 0.59 | 0.56 | 0.89 | 0.85 | 0.85 | **0.91** | 0.90 | 0.90 |
| GSM8K | 0.60 | 0.63 | **0.67** | 0.62 | 0.62 | 0.55 | **0.96** | **0.96** | 0.95 | 0.95 | **0.96** | **0.96** |
| *Avg* | *0.59* | *0.62* | *0.60* | *0.65* | *0.59* | *0.56* | *0.92* | *0.90* | *0.89* | *0.93* | *0.93* | *0.93* |
| **Overall Avg** | 0.70 | 0.70 | 0.69 | 0.70 | **0.71** | 0.66 | 0.89 | 0.87 | 0.87 | **0.90** | **0.90** | **0.90** |

(a) Mathematical tasks

| | AUROC | | | | | | Accuracy | | | | | |
|---|---|---|---|---|---|---|---|---|---|---|---|---|
| Task | VaEl | VeEl | EpEl | VaJu | VeJu | EpJu | VaEl | VeEl | EpEl | VaJu | VeJu | EpJu |
| *gpt20b-high* | | | | | | | | | | | | |
| GPQA Diamond | 0.76 | 0.76 | **0.78** | 0.75 | 0.69 | 0.72 | 0.75 | 0.74 | 0.74 | 0.75 | **0.77** | 0.75 |
| MMLU-Pro Health | **0.74** | 0.73 | **0.74** | 0.72 | 0.73 | 0.67 | 0.74 | 0.74 | 0.74 | 0.74 | **0.75** | **0.75** |
| MMLU-Pro Psychology | **0.73** | 0.72 | **0.73** | 0.71 | 0.71 | 0.65 | **0.74** | 0.73 | **0.74** | **0.74** | **0.74** | 0.73 |
| MMLU-Pro Biology | 0.74 | 0.74 | **0.76** | 0.73 | 0.72 | 0.67 | 0.87 | **0.88** | **0.88** | **0.88** | **0.88** | **0.88** |
| MMLU-Pro Business | **0.83** | 0.82 | **0.83** | 0.82 | 0.82 | 0.75 | 0.85 | 0.84 | 0.85 | **0.86** | 0.85 | 0.85 |
| MMLU-Pro Law | **0.58** | 0.57 | **0.58** | **0.58** | **0.58** | 0.56 | **0.45** | **0.45** | 0.44 | 0.44 | **0.45** | **0.45** |
| *Avg* | *0.73* | *0.72* | *0.74* | *0.72* | *0.71* | *0.67* | *0.73* | *0.73* | *0.73* | *0.74* | *0.74* | *0.74* |
| *deepseek8b* | | | | | | | | | | | | |
| GPQA Diamond | **0.80** | 0.77 | 0.77 | 0.79 | 0.74 | 0.72 | **0.62** | 0.60 | 0.61 | 0.61 | 0.61 | **0.62** |
| MMLU-Pro Health | **0.68** | 0.65 | **0.68** | 0.66 | 0.60 | 0.63 | **0.70** | **0.70** | **0.70** | **0.70** | **0.70** | **0.70** |
| MMLU-Pro Psychology | 0.68 | 0.63 | **0.69** | 0.67 | 0.60 | 0.64 | 0.73 | 0.73 | 0.73 | 0.73 | **0.74** | 0.73 |
| MMLU-Pro Biology | 0.73 | 0.71 | **0.74** | 0.68 | 0.65 | 0.63 | 0.86 | 0.86 | **0.87** | 0.86 | 0.86 | **0.87** |
| MMLU-Pro Business | **0.80** | 0.79 | 0.79 | 0.78 | 0.75 | 0.72 | 0.82 | 0.81 | 0.81 | **0.82** | **0.82** | **0.82** |
| MMLU-Pro Law | **0.58** | 0.56 | 0.57 | **0.58** | 0.55 | 0.56 | **0.44** | 0.43 | 0.43 | 0.43 | 0.43 | 0.43 |
| *Avg* | *0.71* | *0.68* | *0.71* | *0.69* | *0.65* | *0.65* | *0.70* | *0.69* | *0.69* | *0.69* | *0.69* | *0.70* |
| *qwen30b* | | | | | | | | | | | | |
| GPQA Diamond | 0.56 | **0.70** | 0.67 | 0.59 | 0.68 | 0.59 | 0.71 | 0.71 | 0.70 | **0.72** | 0.71 | 0.70 |
| MMLU-Pro Health | 0.57 | **0.68** | 0.67 | 0.60 | 0.66 | 0.59 | 0.76 | 0.76 | 0.75 | **0.77** | 0.76 | **0.77** |
| MMLU-Pro Psychology | 0.55 | 0.65 | **0.66** | 0.61 | 0.64 | 0.56 | 0.77 | 0.77 | 0.77 | **0.78** | **0.78** | **0.78** |
| MMLU-Pro Biology | 0.55 | **0.68** | 0.67 | 0.65 | 0.66 | 0.56 | 0.89 | 0.89 | 0.89 | **0.90** | 0.89 | 0.89 |
| MMLU-Pro Business | 0.62 | **0.76** | **0.76** | 0.67 | 0.72 | 0.64 | 0.86 | 0.85 | 0.85 | **0.86** | **0.86** | **0.86** |
| MMLU-Pro Law | 0.51 | 0.56 | 0.56 | 0.53 | **0.57** | 0.52 | 0.55 | 0.55 | 0.54 | 0.55 | **0.56** | **0.56** |
| *Avg* | *0.56* | *0.67* | *0.66* | *0.61* | *0.66* | *0.58* | *0.76* | *0.76* | *0.75* | *0.76* | *0.76* | *0.76* |
| **Overall Avg** | 0.67 | 0.69 | **0.70** | 0.67 | 0.67 | 0.63 | **0.73** | **0.73** | 0.72 | **0.73** | **0.73** | **0.73** |

(b) Non-mathematical natural and social science tasks

Table 2: Performance across tasks. AUROC (blue) measures the discriminative power of confidence, while Accuracy (violet) measures correctness. Best values per task and metric are bolded (ties all bold). In math, results are highly *model- and dataset-dependent* (non-vanilla elicitation can reduce accuracy; EpJu underperforms in AUROC; no clear dominator among VaEl, VaJu, VeJu; VaEl is a competitive baseline). In non-math, accuracy is stable across methods; epistemic elicitation (EpEl) consistently increases AUROC by roughly 3 points on average, making it a reliable drop-in upgrade.

are bootstrap means over 1,000 resamples over 20 repeats per question (detailed experimental setup described in Section 4).

**Mathematical tasks.** Table 2a reports results on mathematical tasks. In terms of **accuracy**, non-vanilla elicitation (VeEl, EpEl) consistently underperforms on *qwen30b* and *deepseek8b* (loosing 1-3 pts), while only *gpt20b-high* remains robust to the elicitation method. One interpretation is

that math is in-distribution for RLVR training, so additional steering instructions may conflict with reasoning strategies the models have already internalized. A simpler one is that eliciting in the same iteration is taking resources from solving reasoning-heavy mathematical tasks. In terms of **AUROC**, Epistemic Markers Judge (EpJu) performs consistently worst, suggesting that epistemic markers are not reliable uncertainty cues or are not well exploited by the judge. After removing these weak options, the viable contenders are *vanilla elicitation* (VaEl) and the two judge variants (VaJu, VeJu). Their relative performance is highly model- and dataset-dependent: VeJu leads on *gpt20b-high* but loses to VaEl by 6 AUROC percentage points on AIME 2025; VaEl is strongest on *deepseek8b*; and VaJu leads on *qwen30b*. On average, these wins cancel out, leaving no consistent advantage. **Takeaway:** Some methods clearly underperform (VeEl, EpEl, EpJu), while among the rest, there is no universally superior choice. For these methods, performance is heterogeneous and depends on the specific model and dataset.

**Non-mathematical tasks.** Table 2b reports results on non-mathematical tasks. **Accuracy** remains stable across all methods, with no systematic drops beyond 1 percentage point. In terms of **AUROC**, epistemic elicitation (EpEl) provides a consistent improvement of about **3 points** over vanilla elicitation and judge variants, while verification elicitation is only slightly weaker. This gain contrasts with mathematics tasks, but is consistent with non-math domains being out-of-distribution for RLVR training: models are more steerable by richer uncertainty instructions, which enhance discriminative power without harming correctness. Judge methods now consistently underperform, likely because reasoning chains in non-math domains provide less explicit structure for a post-hoc judge to exploit. **Takeaway:** EpEl is a reliable drop-in upgrade over VaEl for non-math tasks, delivering consistent AUROC gains with no accuracy cost. Judges fall behind in confidence quality and offer no accuracy improvements to justify the extra LRM calls.

**Summary.** The confidence estimation results in LRMs show a sharp contrast between domains. In **math** tasks, several methods clearly underperform (VeEl, EpEl, EpJu), and among the rest, performance varies across models and datasets with no universally superior choice. In **non-math** tasks, the pattern is more stable: EpEl consistently improves AUROC by about 3 points without harming accuracy, while judge methods add cost without benefit. This suggests that in-distribution domains (like math), resist further elicitation prompt engineering but can selectively benefit from a separate LLM confidence judge, whereas out-of-distribution domains gain from richer uncertainty instructions.

## 5.2 EVALUATING SELF-CONSISTENCY AND VCSC

We evaluate self-consistency (SC), average verbalized confidence ($VC_{avg}$), and their hybrid (VCSC), as described in Section 3.2. For verbalized confidence, we use *vanilla elicitation* (VaEl) on mathematical tasks and *epistemic elicitation* (EpEl) on non-mathematical tasks, reflecting the insights from Section 5.1. To balance coverage while keeping analysis focused, we select two math and two non-math tasks: for math, `AIME 2024/2025`, which remain unsolved and are the standard choice in recent literature; for non-math, `GPQA Diamond`, which spans multiple natural science domains, and `MMLU-Pro Psychology`, a social science domain that involves both knowledge and reasoning and is especially critical for safety-sensitive applications such as mental health AI. We report the accuracy and AUROC, computed as bootstrap means over 1,000 resamples using up to 100 repeats per question for reliable subsampling at higher $K$ (see Section 4).

**Accuracy.** Across models, accuracy improves modestly with majority voting, rising by about 3–4 points and saturating by $K \approx 8$. Importantly, there is *no gain from $K = 1$ to $K = 2$*, meaning that all AUROC improvements discussed below occur under the same accuracy conditions having an equally sized set of correct vs. incorrect answers to descriminate.

**Self-consistency (SC).** Table 3 shows that SC displays heterogeneous AUROC behavior. On *deepseek8b* and *qwen30b*, AUROC improves steadily with more samples ($\sim 0.72$ at $K = 2$ to $\sim 0.85$ at $K = 64$). In contrast, on *gpt20b-high* AUROC peaks already at $K = 2$ (0.65) and then declines with larger $K$. This suggests that in LRMs, where single traces already branch internally, additional sampling can either strengthen the signal or blur it, depending on the model. SC therefore remains informative, but it is fragile and highly model-dependent.

**Average verbalized confidence ($VC_{avg}$).** In contrast, $VC_{avg}$ improves smoothly and monotonically with $K$ across all models, from an average of 0.73 at $K = 1$ to 0.83 at $K = 8$ and 0.86 at $K = 64$.

Notably, on *deepseek8b* AIME datasets, $VC_{avg}$ reaches near-perfect AUROC (~1.0) even though accuracy plateaus far lower (~0.83).

**Hybrid (VCSC).** Combining SC and $VC_{avg}$ (VCSC, $\lambda = 0.5$) delivers striking gains at small budgets. With only two repeats, VCSC reaches 0.85 AUROC, outperforming $VC_{avg}$ at the same budget by 10 points overall and up to +19 points on *qwen30b*. Put differently, $VC_{avg}$ requires ~32 samples to match VCSC with just 2, while SC lags behind by 9 points even with 64 traces. This demonstrates that SC and VC capture complementary signals in reasoning LLMs: behavioral agreement provides leverage exactly when verbalized confidence is still noisy, making the hybrid uniquely effective at low $K$. At larger budgets, however SC become uninformative as we observe with gpt20b on AIME 2025, partially hurting VCSC as a result.

**Summary.** Overall, accuracy gains saturate quickly and cannot account for the large AUROC improvements we observe. SC alone is unreliable in LRMs, but provides a great value when paired with VC. $VC_{avg}$, unlike accuracy, scales well with $K$, highlighting a generation–verification gap: while models do not reliably generate the correct answer even with many repeats, they become very accurate at *detecting when they are wrong*. By aggregating confidence over multiple majority-aligned traces, $VC_{avg}$ amplifies these uncertainty cues, becoming sharper and yielding robust discriminative power even without further accuracy gains. The hybrid VCSC combines these strengths, delivering dramatic AUROC gains with as few as two repeats.

## 5.3 ANALYSIS AND ABLATIONS

In this section, we ablate the reasoning effort, reasoning mode, and vary the mixing weight $\lambda$ to explore the mechanism underlying VCSC beyond our main results.

**Ablating reasoning effort and reasoning mode.** We ablated the reasoning effort (gpt20b-high) and reasoning mode (qwen30b) and present results averaged across tasks in Figure 1 (per-task results are in Figures 4 and 5 in Appendix A). Reducing reasoning lowers accuracy but leaves SC-based confidence either improved (comparing to AIME SC drop we observed in Table 3) or largely intact (qwen). VC is sometimes better in non/low-thinking mode, but generally (high)thinking models are able to maintain high AUROC despite increased accuracy. Importantly, VCSC consistently delivers strong gains, especially at small $K$ across all settings.

| Task | Accuracy K=1 | K=2 | K=8 | K=32 | K=64 | SC K=1 | K=2 | K=8 | K=32 | K=64 | VC avg K=1 | K=2 | K=8 | K=32 | K=64 | VCSC K=1 | K=2 | K=8 | K=32 | K=64 |
|---|---|---|---|---|---|---|---|---|---|---|---|---|---|---|---|---|---|---|---|---|
| | | | | | | | | *gpt20b-high* | | | | | | | | | | | | |
| AIME 2024 | 0.91 | 0.91 | 0.93 | 0.93 | 0.93 | 0.50 | 0.69 | 0.70 | 0.62 | 0.57 | 0.73 | 0.73 | 0.85 | 0.88 | 0.89 | 0.73 | 0.91 | 0.90 | 0.95 | **0.96** |
| AIME 2025 | 0.90 | 0.90 | **0.93** | **0.93** | 0.93 | 0.50 | 0.63 | 0.40 | 0.25 | 0.16 | 0.79 | 0.77 | 0.84 | 0.84 | 0.84 | 0.79 | 0.87 | 0.69 | 0.64 | 0.57 |
| GPQA Diamond | 0.74 | 0.74 | 0.78 | 0.78 | 0.79 | 0.50 | 0.66 | 0.72 | 0.74 | 0.75 | 0.76 | 0.78 | 0.81 | 0.81 | 0.81 | 0.76 | **0.82** | 0.81 | 0.80 | 0.80 |
| MMLU-Pro Psych. | 0.74 | 0.74 | 0.75 | 0.75 | 0.75 | 0.50 | 0.63 | 0.73 | 0.78 | 0.79 | 0.74 | 0.76 | 0.79 | 0.80 | 0.80 | 0.74 | 0.79 | 0.81 | **0.82** | **0.82** |
| *Avg* | *0.82* | *0.82* | ***0.85*** | ***0.85*** | ***0.85*** | *0.50* | *0.65* | *0.64* | *0.60* | *0.57* | *0.76* | *0.76* | *0.82* | *0.83* | *0.84* | *0.76* | ***0.85*** | *0.80* | *0.80* | *0.79* |
| | | | | | | | | *deepseek8b* | | | | | | | | | | | | |
| AIME 2024 | 0.75 | 0.75 | 0.82 | 0.82 | 0.83 | 0.50 | 0.86 | 0.92 | 0.94 | 0.94 | 0.88 | 0.89 | 0.97 | **1.00** | **1.00** | 0.88 | 0.97 | 0.99 | 0.99 | 0.99 |
| AIME 2025 | 0.66 | 0.65 | 0.76 | 0.78 | 0.78 | 0.50 | 0.85 | 0.90 | 0.91 | 0.92 | 0.86 | 0.88 | 0.97 | **1.00** | **1.00** | 0.86 | 0.95 | 0.95 | 0.96 | 0.96 |
| GPQA Diamond | 0.59 | 0.59 | 0.62 | 0.62 | 0.62 | 0.50 | 0.66 | 0.75 | 0.77 | 0.78 | 0.77 | 0.80 | 0.84 | **0.85** | **0.85** | 0.77 | 0.82 | 0.84 | 0.84 | 0.84 |
| MMLU-Pro Psych. | 0.73 | 0.73 | 0.75 | 0.75 | 0.75 | 0.50 | 0.63 | 0.72 | 0.76 | **0.78** | 0.69 | 0.71 | 0.72 | 0.72 | 0.72 | 0.69 | 0.75 | 0.77 | **0.78** | **0.78** |
| *Avg* | *0.68* | *0.68* | *0.74* | *0.74* | *0.74* | *0.50* | *0.75* | *0.82* | *0.84* | *0.86* | *0.80* | *0.82* | *0.88* | ***0.89*** | ***0.89*** | *0.80* | *0.87* | ***0.89*** | ***0.89*** | ***0.89*** |
| | | | | | | | | *qwen30b* | | | | | | | | | | | | |
| AIME 2024 | 0.88 | 0.88 | 0.92 | 0.93 | 0.93 | 0.50 | 0.78 | 0.77 | 0.82 | 0.86 | 0.61 | 0.68 | 0.87 | 0.96 | **0.98** | 0.61 | 0.86 | 0.88 | 0.88 | 0.89 |
| AIME 2025 | 0.84 | 0.83 | 0.89 | 0.90 | 0.90 | 0.50 | 0.86 | 0.94 | 0.96 | 0.96 | 0.59 | 0.65 | 0.82 | 0.89 | 0.92 | 0.59 | 0.90 | 0.95 | 0.96 | **0.97** |
| GPQA Diamond | 0.70 | 0.70 | 0.71 | 0.71 | 0.71 | 0.50 | 0.64 | 0.76 | 0.80 | 0.81 | 0.68 | 0.72 | 0.77 | 0.79 | 0.79 | 0.68 | 0.77 | 0.82 | **0.83** | **0.83** |
| MMLU-Pro Psych. | 0.77 | 0.77 | 0.78 | 0.78 | 0.78 | 0.50 | 0.61 | 0.70 | 0.74 | 0.76 | 0.65 | 0.68 | 0.72 | 0.74 | 0.74 | 0.65 | 0.73 | 0.78 | **0.80** | **0.80** |
| *Avg* | *0.80* | *0.80* | *0.82* | *0.83* | *0.83* | *0.50* | *0.72* | *0.79* | *0.83* | *0.85* | *0.63* | *0.68* | *0.80* | *0.84* | *0.86* | *0.63* | *0.82* | *0.86* | ***0.87*** | ***0.87*** |
| **Overall Avg** | 0.77 | 0.77 | 0.80 | 0.81 | 0.81 | 0.50 | 0.71 | 0.75 | 0.76 | 0.76 | 0.73 | 0.75 | 0.83 | 0.85 | **0.86** | 0.73 | 0.85 | 0.85 | 0.85 | 0.85 |

Table 3: Confidence estimation quality as a function of sample count $K$ across self-consistency (SC), average verbalized confidence ($VC_{avg}$), and their hybrid (VCSC). Reported numbers are AUROC (shaded) and Accuracy (gray). Best values per task and method are bolded (ties all bold). Accuracy saturates by $K \approx 8$, while AUROC shows distinct scaling patterns: SC is fragile and model-dependent, $VC_{avg}$ improves smoothly with $K$, and VCSC delivers the largest early gains, reaching ~10 AUROC points above alternatives at $K = 2$. Results are bootstrap means over 1,000 resamples drawing from a pool of up to 100 repeats per question.

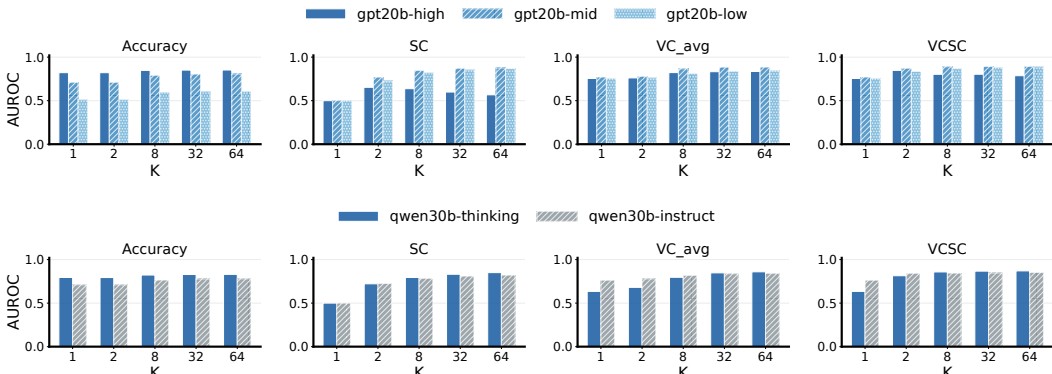

Figure 1: Ablating reasoning effort (gpt20b-high) and mode (qwen30b), averaged across tasks.

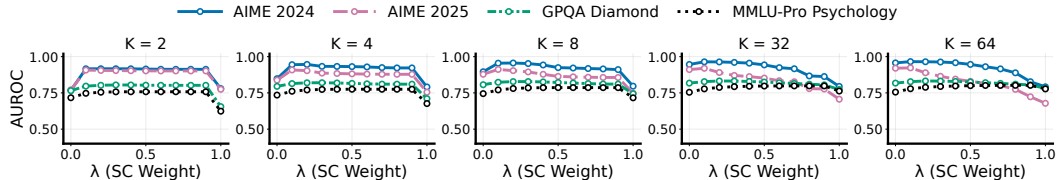

Figure 2: Effect of mixing weight $\lambda$ in VCSC (averaged over models).

**Effect of mixing weight $\lambda$.** We used VCSC $= \lambda\,\text{SC} + (1 - \lambda)\,\text{VC}_{\text{avg}}$, with $\lambda = 0.5$ as a default in Table 3. Endpoints at $\lambda{=}0$ and $\lambda{=}1$ correspond to pure $\text{VC}_{\text{avg}}$ and SC respectively, and are also included in the table. In Figure 2, we explore how intermediate mixes behave across budgets $K$ (per-model curves are in Figure 3 in Appendix A). Two trends emerge. At **low budget** ($K{=}2$), performance is flat across a broad range of $\lambda$, making $0.5$ a safe default. As $K$ grows ($8{\rightarrow}32{\rightarrow}64$), the optimum shifts left toward $\lambda{\approx}0$, with $\text{VC}_{\text{avg}}$ carrying most of the signal. Crucially, at $K{=}8$ and small $\lambda$ (0.1–0.3), VCSC *matches* the best AUROC of $\text{VC}_{\text{avg}}$ at $K{=}64$, cutting sampling cost by up to **8**$\times$. Thus, while $\lambda{=}0.5$ already delivers strong small-$K$ gains without tuning, light adjustment (or decaying $\lambda$ as $K$ increases) can recover nearly all large-$K$ performance at much lower budgets. The effect is strongest on math tasks, while non-math remains less sensitive to this parameter.

Overall, ablation results suggest that the gains offered by VCSC are not specific to LRMs as, for example, jump from K=1 to K=2, is at least as steep for non-reasoning LLMs. This means that despite reasoning models operating on a different task competence (accuracy) level, they are able to maintain high confidence quality levels and scale with repeats just as their non-thinking counterparts do. The results can be further improved by selecting the optimal values $\lambda$.

## 6 CONCLUSION

We presented the first systematic study of uncertainty estimation in reasoning LLMs going beyond vanilla verbalized confidence. Across nine tasks and three LRMs, we found that advanced VC methods rarely outperform vanilla elicitation on math tasks, while epistemic elicitation yields modest but consistent gains on non-math tasks. Most importantly, we showed that combining VC with self-consistency (VCSC) delivers dramatic confidence estimation improvements: over 10 AUROC points on average with just two repetitions, and near-perfect discrimination on some task-model combinations.

These findings establish VCSC as an overlooked yet highly effective approach for uncertainty estimation in reasoning (and non-reasoning) LLMs. Beyond this immediate practical recipe, our results deepen understanding of how LRMs expose, and can be made to exploit, their own uncertainty.

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

APPENDIX

## A  ADDITIONAL RESULTS

This Section introduces additional results to the main results.

Tables 4 and 6 have results on mathematics tasks, and Tables 5 and 7 other sciences tasks on ECE and NLL measures.

Tables 8 and 9 show results for ECE and NLL for VCSC across models and different K-values.

| Task | VaEl | VeEl | EpEl | VaJu | VeJu | EpJu |
|------|------|------|------|------|------|------|
| *gpt20b-high* | | | | | | |
| AIME 2024 | 0.04 | **0.03** | **0.03** | 0.05 | 0.04 | 0.05 |
| AIME 2025 | **0.03** | 0.04 | **0.03** | **0.03** | 0.04 | 0.04 |
| GSM8K | 0.05 | 0.04 | 0.05 | 0.04 | **0.03** | 0.04 |
| *Avg* | *0.04* | *0.04* | *0.04* | *0.04* | *0.04* | *0.04* |
| *deepseek8b* | | | | | | |
| AIME 2024 | 0.14 | 0.14 | 0.16 | 0.13 | **0.10** | 0.13 |
| AIME 2025 | 0.20 | 0.22 | 0.22 | 0.18 | **0.12** | 0.17 |
| GSM8K | **0.04** | 0.05 | 0.05 | 0.05 | 0.05 | 0.06 |
| *Avg* | *0.13* | *0.14* | *0.14* | *0.12* | *0.09* | *0.12* |
| *qwen30b* | | | | | | |
| AIME 2024 | 0.06 | 0.07 | 0.08 | **0.03** | 0.05 | 0.08 |
| AIME 2025 | 0.07 | 0.12 | 0.11 | **0.06** | 0.07 | 0.09 |
| GSM8K | **0.01** | 0.04 | 0.04 | 0.02 | 0.04 | 0.06 |
| *Avg* | *0.05* | *0.08* | *0.08* | *0.04* | *0.05* | *0.08* |
| **Overall Avg** | 0.07 | 0.09 | 0.09 | 0.07 | **0.06** | 0.08 |

Table 4: Math (ECE).

| Task | VaEl | VeEl | EpEl | VaJu | VeJu | EpJu |
|------|------|------|------|------|------|------|
| *gpt20b-high* | | | | | | |
| GPQA Diamond | 0.18 | **0.15** | 0.16 | 0.19 | **0.15** | 0.17 |
| MMLU-Pro Health | 0.19 | 0.17 | 0.17 | 0.20 | **0.16** | 0.19 |
| MMLU-Pro Psychology | 0.20 | **0.17** | 0.19 | 0.20 | **0.17** | 0.21 |
| MMLU-Pro Biology | 0.08 | **0.05** | 0.07 | 0.08 | **0.05** | 0.07 |
| MMLU-Pro Business | 0.10 | 0.08 | 0.09 | 0.10 | **0.07** | 0.09 |
| MMLU-Pro Law | 0.48 | **0.44** | 0.46 | 0.48 | **0.44** | 0.47 |
| *Avg* | *0.20* | *0.18* | *0.19* | *0.21* | *0.17* | *0.20* |
| *deepseek8b* | | | | | | |
| GPQA Diamond | 0.22 | 0.18 | 0.19 | 0.26 | **0.15** | 0.22 |
| MMLU-Pro Health | 0.21 | 0.15 | 0.18 | 0.22 | **0.12** | 0.18 |
| MMLU-Pro Psychology | 0.18 | 0.12 | 0.16 | 0.19 | **0.08** | 0.16 |
| MMLU-Pro Biology | 0.06 | **0.03** | 0.07 | 0.08 | 0.04 | 0.07 |
| MMLU-Pro Business | 0.09 | 0.07 | 0.09 | 0.11 | **0.04** | 0.08 |
| MMLU-Pro Law | 0.44 | 0.39 | 0.40 | 0.46 | **0.38** | 0.44 |
| *Avg* | *0.20* | *0.16* | *0.18* | *0.22* | *0.14* | *0.19* |
| *qwen30b* | | | | | | |
| GPQA Diamond | 0.23 | **0.22** | **0.22** | 0.24 | **0.22** | 0.26 |
| MMLU-Pro Health | **0.19** | 0.20 | 0.20 | **0.19** | 0.20 | 0.20 |
| MMLU-Pro Psychology | **0.17** | 0.19 | 0.19 | 0.18 | 0.18 | 0.19 |
| MMLU-Pro Biology | **0.06** | 0.08 | 0.08 | 0.07 | 0.08 | 0.10 |
| MMLU-Pro Business | **0.09** | 0.11 | 0.10 | 0.10 | 0.10 | 0.10 |
| MMLU-Pro Law | 0.40 | **0.39** | 0.40 | 0.40 | **0.39** | 0.43 |
| *Avg* | *0.19* | *0.20* | *0.20* | *0.20* | *0.19* | *0.21* |
| **Overall Avg** | 0.20 | 0.18 | 0.19 | 0.21 | **0.17** | 0.20 |

Table 5: Other Sciences (ECE).

| Task | VaEl | VeEl | EpEl | VaJu | VeJu | EpJu |
|------|------|------|------|------|------|------|
| *gpt20b-high* | | | | | | |
| AIME 2024 | 0.38 | 0.39 | 0.37 | 0.40 | **0.26** | 0.46 |
| AIME 2025 | 0.29 | 0.37 | 0.42 | 0.29 | **0.28** | 0.38 |
| GSM8K | 0.43 | 0.49 | 0.61 | **0.31** | 0.32 | 0.54 |
| *Avg* | *0.37* | *0.42* | *0.47* | *0.33* | ***0.29*** | *0.46* |
| *deepseek8b* | | | | | | |
| AIME 2024 | 0.65 | 1.06 | 1.32 | 0.76 | **0.58** | 1.09 |
| AIME 2025 | 0.82 | 1.58 | 1.73 | 0.90 | **0.73** | 1.63 |
| GSM8K | **0.52** | 0.77 | 0.79 | 0.56 | 0.52 | 0.62 |
| *Avg* | *0.66* | *1.14* | *1.28* | *0.74* | ***0.61*** | *1.11* |
| *qwen30b* | | | | | | |
| AIME 2024 | 0.35 | 0.53 | 0.54 | **0.28** | 0.60 | 1.17 |
| AIME 2025 | 0.45 | 0.89 | 0.82 | **0.35** | 0.65 | 1.46 |
| GSM8K | **0.23** | 0.60 | 0.51 | 0.32 | 0.61 | 0.67 |
| *Avg* | *0.34* | *0.67* | *0.62* | ***0.32*** | *0.62* | *1.10* |
| **Overall Avg** | **0.46** | 0.74 | 0.79 | **0.46** | 0.51 | 0.89 |

Table 6: Math (NLL).

| Task | VaEl | VeEl | EpEl | VaJu | VeJu | EpJu |
|------|------|------|------|------|------|------|
| *gpt20b-high* | | | | | | |
| GPQA Diamond | 0.65 | **0.58** | 0.60 | 0.84 | 0.85 | 0.87 |
| MMLU-Pro Health | **0.72** | 0.75 | 0.90 | 0.88 | **0.72** | 1.43 |
| MMLU-Pro Psychology | 0.74 | 0.74 | 1.01 | 0.87 | **0.73** | 1.60 |
| MMLU-Pro Biology | **0.42** | 0.44 | 0.52 | 0.46 | **0.42** | 0.77 |
| MMLU-Pro Business | 0.44 | 0.47 | 0.56 | 0.48 | **0.43** | 0.72 |
| MMLU-Pro Law | 1.51 | **1.33** | 1.59 | 1.84 | 1.49 | 2.50 |
| *Avg* | *0.75* | ***0.72*** | *0.86* | *0.90* | *0.77* | *1.32* |
| *deepseek8b* | | | | | | |
| GPQA Diamond | **0.70** | 0.85 | 1.03 | 0.92 | 0.77 | 1.51 |
| MMLU-Pro Health | **0.78** | 0.99 | 1.52 | 1.06 | 0.79 | 1.95 |
| MMLU-Pro Psychology | 0.70 | 0.85 | 1.27 | 0.93 | **0.68** | 1.83 |
| MMLU-Pro Biology | **0.40** | 0.58 | 0.80 | 0.52 | 0.44 | 1.07 |
| MMLU-Pro Business | **0.48** | 0.75 | 0.90 | 0.64 | 0.54 | 1.10 |
| MMLU-Pro Law | 1.26 | **1.21** | 1.42 | 1.59 | 1.24 | 3.79 |
| *Avg* | *0.72* | *0.87* | *1.16* | *0.94* | *0.74* | *1.88* |
| *qwen30b* | | | | | | |
| GPQA Diamond | **0.86** | 0.90 | 0.87 | 0.88 | 1.33 | 3.89 |
| MMLU-Pro Health | **0.75** | 1.56 | 1.66 | 0.85 | 1.99 | 3.18 |
| MMLU-Pro Psychology | **0.70** | 1.68 | 1.58 | 0.84 | 1.96 | 3.24 |
| MMLU-Pro Biology | **0.37** | 0.82 | 0.84 | 0.42 | 0.95 | 1.60 |
| MMLU-Pro Business | **0.45** | 0.80 | 0.71 | 0.49 | 1.00 | 1.65 |
| MMLU-Pro Law | **1.36** | 2.00 | 1.77 | 1.44 | 3.48 | 7.26 |
| *Avg* | *0.75* | *1.29* | *1.24* | *0.82* | *1.78* | *3.47* |
| **Overall Avg** | **0.74** | 0.96 | 1.09 | 0.89 | 1.10 | 2.22 |

Table 7: Other Sciences (NLL).

Figure 3, 4 and 5 show effect of mixing weight $\lambda$, ablating reasoning effort and ablating reasoning mode respectively. These results are presented separately for each task.

| Task | K=1 | K=2 | K=3 | K=4 | K=5 | K=6 | K=7 | K=8 | K=9 | K=10 | K=15 | K=20 | K=30 | K=40 | K=50 | K=70 |
|---|---|---|---|---|---|---|---|---|---|---|---|---|---|---|---|---|
| | | | | | | | *gpt20b-high* | | | | | | | | | |
| AIME 2024 | 0.08 | 0.06 | 0.05 | 0.06 | **0.04** | **0.04** | **0.04** | **0.04** | **0.04** | **0.04** | **0.04** | **0.04** | **0.04** | **0.04** | **0.04** | **0.04** |
| AIME 2025 | 0.09 | 0.08 | 0.08 | 0.08 | 0.07 | 0.06 | 0.06 | 0.06 | 0.06 | 0.06 | 0.06 | 0.06 | 0.06 | **0.05** | **0.05** | **0.05** |
| GPQA Diamond | 0.21 | 0.16 | 0.14 | 0.13 | 0.13 | 0.12 | 0.12 | 0.12 | 0.11 | 0.11 | 0.11 | 0.11 | **0.10** | **0.10** | **0.10** | **0.10** |
| MMLU-Pro Psychology | 0.23 | 0.19 | 0.18 | 0.18 | 0.17 | 0.17 | 0.17 | 0.17 | 0.17 | 0.17 | 0.17 | **0.16** | **0.16** | **0.16** | **0.16** | **0.16** |
| *Avg* | *0.15* | *0.12* | *0.11* | *0.11* | *0.10* | *0.10* | *0.10* | *0.10* | *0.10* | *0.10* | *0.10* | ***0.09*** | ***0.09*** | ***0.09*** | ***0.09*** | ***0.09*** |
| | | | | | | | *deepseek8b* | | | | | | | | | |
| AIME 2024 | 0.20 | 0.15 | 0.13 | **0.12** | **0.12** | 0.13 | **0.12** | **0.12** | **0.12** | **0.12** | **0.12** | **0.12** | **0.12** | **0.12** | **0.12** | **0.12** |
| AIME 2025 | 0.29 | 0.21 | 0.16 | 0.14 | 0.14 | 0.14 | 0.13 | 0.13 | 0.13 | 0.13 | 0.12 | 0.12 | 0.12 | 0.12 | **0.11** | **0.11** |
| GPQA Diamond | 0.30 | 0.23 | 0.21 | 0.20 | 0.19 | 0.18 | 0.18 | 0.18 | 0.18 | 0.17 | 0.17 | **0.16** | **0.16** | **0.16** | **0.16** | **0.16** |
| MMLU-Pro Psychology | 0.21 | 0.18 | 0.17 | 0.16 | 0.16 | 0.16 | **0.15** | **0.15** | **0.15** | **0.15** | **0.15** | **0.15** | **0.15** | **0.15** | **0.15** | **0.15** |
| *Avg* | *0.25* | *0.19* | *0.17* | *0.16* | *0.15* | *0.15* | ***0.14*** | ***0.14*** | ***0.14*** | ***0.14*** | ***0.14*** | ***0.14*** | ***0.14*** | ***0.14*** | ***0.14*** | ***0.14*** |
| | | | | | | | *qwen30b* | | | | | | | | | |
| AIME 2024 | 0.10 | 0.08 | 0.06 | 0.06 | 0.06 | **0.05** | **0.05** | 0.06 | **0.05** | **0.05** | 0.06 | 0.06 | 0.06 | 0.06 | 0.06 | 0.06 |
| AIME 2025 | 0.14 | 0.11 | 0.09 | 0.09 | 0.09 | 0.08 | 0.08 | 0.08 | 0.08 | 0.08 | 0.08 | 0.08 | 0.08 | 0.08 | 0.08 | 0.07 |
| GPQA Diamond | 0.26 | 0.22 | 0.21 | 0.20 | 0.20 | 0.20 | 0.20 | 0.20 | 0.20 | 0.19 | 0.19 | 0.19 | 0.19 | 0.19 | 0.19 | 0.19 |
| MMLU-Pro Psychology | 0.21 | 0.19 | 0.18 | 0.18 | 0.18 | 0.18 | 0.18 | **0.17** | **0.17** | **0.17** | **0.17** | **0.17** | **0.17** | **0.17** | **0.17** | **0.17** |
| *Avg* | *0.18* | *0.15* | *0.14* | *0.13* | *0.13* | *0.13* | *0.13* | *0.13* | ***0.12*** | ***0.12*** | ***0.12*** | ***0.12*** | ***0.12*** | ***0.12*** | ***0.12*** | ***0.12*** |
| **Overall Avg** | 0.19 | 0.15 | 0.14 | 0.13 | 0.13 | 0.13 | **0.12** | **0.12** | **0.12** | **0.12** | **0.12** | **0.12** | **0.12** | **0.12** | **0.12** | **0.12** |

Table 8: ECE results for VCSC across models and Ks.

| Task | K=1 | K=2 | K=3 | K=4 | K=5 | K=6 | K=7 | K=8 | K=9 | K=10 | K=15 | K=20 | K=30 | K=40 | K=50 | K=70 |
|---|---|---|---|---|---|---|---|---|---|---|---|---|---|---|---|---|
| | | | | | | | *gpt20b-high* | | | | | | | | | |
| AIME 2024 | 0.55 | 0.24 | 0.19 | 0.18 | 0.18 | 0.18 | 0.18 | 0.18 | **0.17** | **0.17** | **0.17** | **0.17** | **0.17** | **0.17** | **0.17** | **0.17** |
| AIME 2025 | 0.55 | 0.32 | 0.29 | 0.29 | **0.28** | **0.28** | **0.28** | **0.28** | **0.28** | **0.28** | **0.28** | **0.28** | **0.28** | **0.28** | **0.28** | **0.28** |
| GPQA Diamond | 0.79 | 0.59 | 0.54 | 0.52 | 0.50 | 0.50 | 0.49 | 0.49 | 0.48 | 0.48 | 0.48 | **0.47** | **0.47** | **0.47** | **0.47** | **0.47** |
| MMLU-Pro Psychology | 1.13 | 0.80 | 0.71 | 0.68 | 0.66 | 0.64 | 0.63 | 0.62 | 0.61 | 0.61 | 0.59 | 0.59 | 0.58 | 0.58 | **0.57** | **0.57** |
| *Avg* | *0.76* | *0.49* | *0.43* | *0.42* | *0.40* | *0.40* | *0.40* | *0.39* | *0.38* | *0.38* | *0.38* | *0.38* | *0.38* | *0.38* | ***0.37*** | ***0.37*** |
| | | | | | | | *deepseek8b* | | | | | | | | | |
| AIME 2024 | 0.86 | 0.32 | 0.25 | 0.23 | 0.22 | 0.22 | 0.21 | 0.21 | 0.20 | 0.20 | 0.20 | **0.19** | **0.19** | **0.19** | **0.19** | **0.19** |
| AIME 2025 | 1.19 | 0.49 | 0.38 | 0.35 | 0.33 | 0.32 | 0.31 | 0.31 | 0.31 | 0.30 | 0.30 | **0.29** | **0.29** | **0.29** | **0.29** | **0.29** |
| GPQA Diamond | 1.21 | 0.74 | 0.66 | 0.62 | 0.60 | 0.59 | 0.58 | 0.58 | 0.57 | 0.57 | 0.56 | 0.56 | **0.55** | **0.55** | **0.55** | **0.55** |
| MMLU-Pro Psychology | 1.40 | 0.94 | 0.81 | 0.75 | 0.72 | 0.69 | 0.68 | 0.66 | 0.65 | 0.64 | 0.62 | 0.61 | 0.60 | 0.60 | **0.59** | **0.59** |
| *Avg* | *1.16* | *0.62* | *0.52* | *0.49* | *0.47* | *0.45* | *0.45* | *0.44* | *0.43* | *0.43* | *0.42* | *0.41* | *0.41* | *0.41* | ***0.40*** | ***0.40*** |
| | | | | | | | *qwen30b* | | | | | | | | | |
| AIME 2024 | 0.51 | 0.30 | 0.26 | 0.24 | 0.23 | 0.23 | 0.22 | 0.22 | 0.22 | 0.21 | 0.21 | 0.20 | 0.20 | 0.20 | **0.19** | **0.19** |
| AIME 2025 | 0.70 | 0.34 | 0.27 | 0.24 | 0.23 | 0.22 | 0.21 | 0.21 | 0.21 | 0.21 | 0.20 | 0.20 | **0.19** | **0.19** | **0.19** | **0.19** |
| GPQA Diamond | 1.10 | 0.82 | 0.75 | 0.71 | 0.69 | 0.67 | 0.66 | 0.66 | 0.65 | 0.65 | 0.64 | 0.63 | 0.63 | **0.62** | **0.62** | **0.62** |
| MMLU-Pro Psychology | 1.75 | 1.13 | 0.94 | 0.85 | 0.80 | 0.77 | 0.75 | 0.73 | 0.72 | 0.71 | 0.68 | 0.67 | 0.66 | **0.65** | **0.65** | **0.65** |
| *Avg* | *1.02* | *0.65* | *0.55* | *0.51* | *0.49* | *0.47* | *0.46* | *0.46* | *0.45* | *0.44* | *0.43* | *0.43* | *0.42* | *0.42* | ***0.41*** | ***0.41*** |
| **Overall Avg** | 0.98 | 0.59 | 0.50 | 0.47 | 0.45 | 0.44 | 0.44 | 0.43 | 0.42 | 0.42 | 0.41 | 0.41 | 0.40 | 0.40 | **0.39** | **0.39** |

Table 9: NLL results for VCSC across models and Ks.

## B  DETAILED PROMPTS

This Section describes the detailed prompts, and Figure 6 provides an overview of those used to obtain answers and elicit confidence.

Next are given exact prompt descriptions. Prompt 1 is used for getting model-elicited uncertainties. Prompt 2 is used to get the LRM thought trace without uncertainties, and after prompt 3 is used as a judge, giving us the Judge method. Prompts 1, 2 and 3 are used for multiple-answer questions. Prompts 8, 9 and 10 are like prompts 1, 2 and 3, but for the math dataset.

Inside the prompts 1, 2 and 3 (same for 8, 9 and 10), there are brackets which are used for inputting variables. *Question*, *choices*, and *letter* correspond to the question in hand from the dataset, but

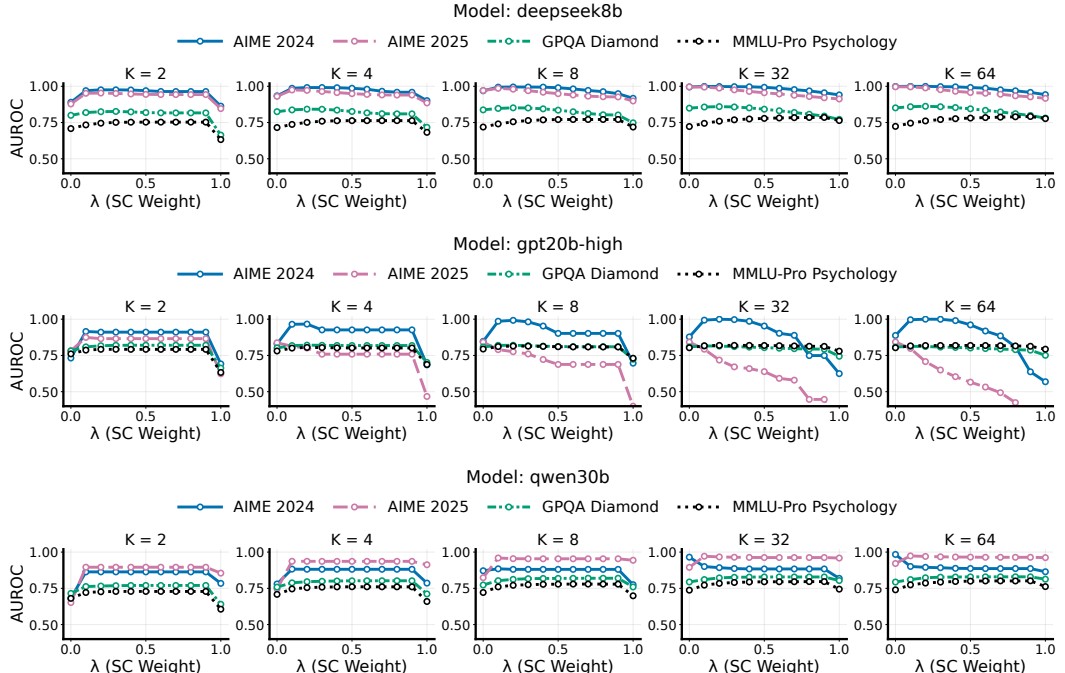

Figure 3: Effect of mixing weight $\lambda$ in VCSC. Shown separately for each model.

```
You are given a multiple choice question.

**Solve the problem**, showing your reasoning step by step. After solving, provide your
    confidence in your answer.

{uncertainty_instructions}

{question}

{choices}

Your response must *end* with exactly two lines of the *exact* format below (no quotes) as the
    very final lines of your answer:
'ANSWER: $LETTER'
'CONFIDENCE: $NUMBER'

- $LETTER must be one of the following options: {letters}

Think step by step before answering and show your reasoning first.
```

Listing 1: LRM prompt for multiple choice question confidence elicitation

*uncertainty_instructions* are subprompts which are based on the method type: vanilla (prompt 4), verification (prompt 5) and epistemic for the elicitation and the judge method (prompt 6 and 7).

```
You are given a multiple choice question.

Solve the problem, showing your reasoning step by step.

{question}

{choices}

Your response must *end* with exactly one line of the *exact* format below (no quotes) as the
    very final line of your answer:
'ANSWER: $LETTER'

- $LETTER must be one of the following options: {letters}

Think step by step before answering and show your reasoning first.
```

Listing 2: LRM prompt for multiple choice question without uncertainty elicitation

```
You are given a problem along with a solver's full reasoning process and the answer they
    arrived at.

{uncertainty_instructions}

[BEGIN PROBLEM]
{question}
[END PROBLEM]

[BEGIN REASONING AND ANSWER]
{reasoning_and_answer}
[END REASONING AND ANSWER]

Think step by step. End your final answer with this exact format as the final line:
CONFIDENCE: [number between 1 and 100]
```

Listing 3: LRM prompt for multiple choice question for judging solver's reasoning process

```
Give a confidence number from 1 to 100 that represents your overall confidence that the **
    final answer is correct**.
```

Listing 4: Vanilla uncertainty prompt for uncertainty instructions bracket in the main prompt

```
Give a confidence number from 1 to 100 that represents how likely the **final answer is
    correct**, *based on the quality of the reasoning*. Focus on soundness, validity, and
    coherence. Pay attention to errors, unsupported steps, flawed assumptions, etc.

    1   - Reasoning fatally flawed; **final answer is almost certainly wrong**
    25  - Major gaps or errors; **final answer is probably wrong**
    50  - Mixed support; **final answer is plausible but not well-proved**
    75  - Strong reasoning with only minor issues; **final answer is probably right**
    100 - Airtight reasoning; **final answer is correct**
```

Listing 5: Verification uncertainty prompt for uncertainty instructions bracket in the main prompt

```
Give a confidence number from 1 to 100 that represents **how confident you are in your final
    answer**.
Treat your final answer as fixed and do not try to re-solve the problem. Focus primarily on
    your expressed certainty or doubt-hedging, hesitation, self-correction.

    1   - You feel like you're making a random guess; openly unsure
    25  - You have significant doubts or hesitations
    50  - Mixed feelings; some confidence, some doubt
    75  - Mostly confident with minor reservations
    100 - Completely certain; no doubts whatsoever
```

Listing 6: Epistemic uncertainty prompt for uncertainty instructions bracket in the main prompt for elicitation

```
Give a confidence number from 1 to 100 that represents **how confident the solver is in their
    final answer**.
Treat their final answer as fixed and do not try to re-solve the problem. Focus primarily on
    their expressed certainty or doubt-hedging, hesitation, self-correction.

    1   - They feel like they're making a random guess; openly unsure
    25  - They have significant doubts or hesitations
    50  - Mixed feelings; some confidence, some doubt
    75  - Mostly confident with minor reservations
    100 - Completely certain; no doubts whatsoever
```

Listing 7: Epistemic uncertainty prompt for uncertainty instructions bracket in the main prompt for judge

```
You are given a math problem.

**Solve the problem**, showing your reasoning step by step. After solving, provide your
    confidence in your answer.

{uncertainty_instructions}

{prompt}

Your response must *end* with exactly two lines of the *exact* format below (no quotes) as the
    very final lines of your answer:
'ANSWER: $ANSWER'
'CONFIDENCE: $NUMBER'

- Do not use LaTeX boxes like \boxed in the final lines; output plain text only.
- Think step by step before answering and show your reasoning first.
```

Listing 8: LRM prompt for math question confidence elicitation

```
You are given a math problem.

Solve the problem, showing your reasoning step by step.

{prompt}

Your response must *end* with exactly one line of the *exact* format below (no quotes) as the
    very final line of your answer:
'ANSWER: $ANSWER'

- Do not use LaTeX boxes like \boxed in the final line; output plain text only.
- Think step by step before answering and show your reasoning first.
```

Listing 9: LRM prompt for math question without uncertainty elicitation

```
You are given a problem along with a solver's full reasoning process and the answer they
    arrived at.

{uncertainty_instructions}

[BEGIN PROBLEM]
{question}
[END PROBLEM]

[BEGIN REASONING AND ANSWER]
{reasoning_and_answer}
[END REASONING AND ANSWER]

Think step by step. End your final answer with this exact format as the final line:
CONFIDENCE: [number between 1 and 100]

- Do not use LaTeX boxes like \boxed in the final line; output plain text only.
```

Listing 10: LRM prompt for math question for judging solver's reasoning process

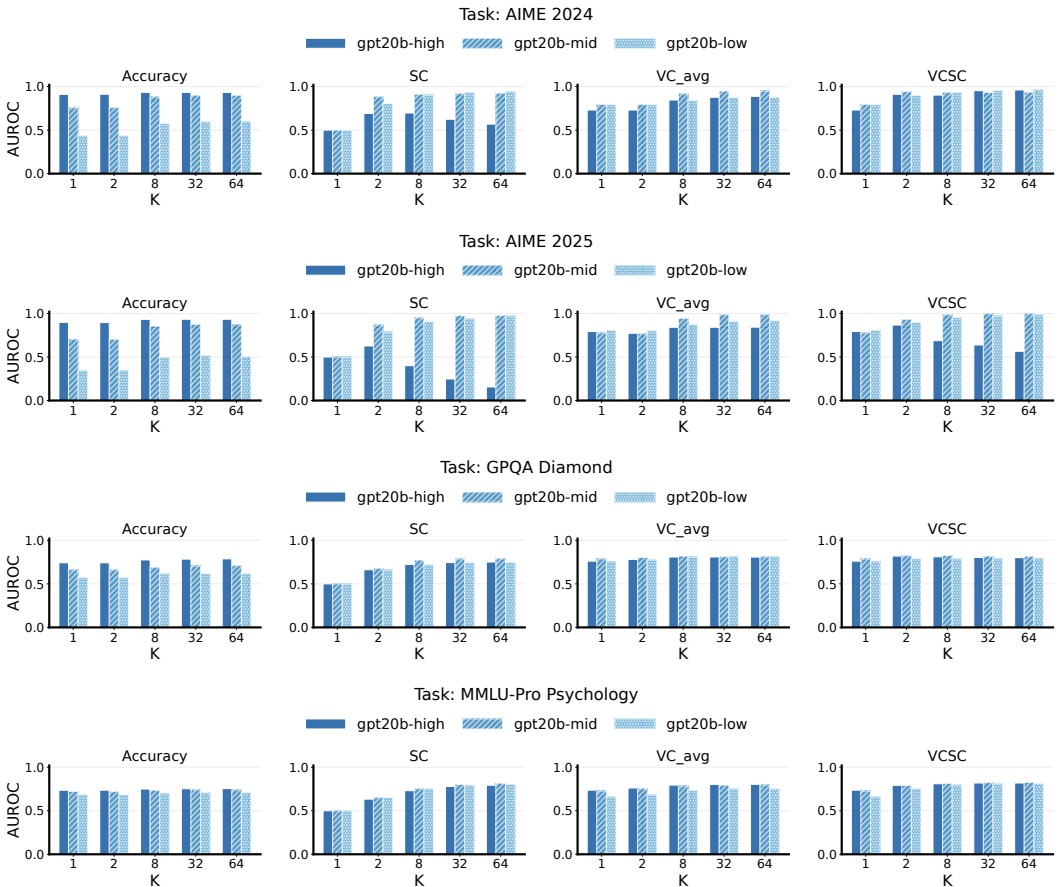

Figure 4: Ablating reasoning effort. Shown separately for each task.

# LLM USAGE

In preparing this paper, we used a large language model (ChatGPT) as a writing assistant. Its role was limited to polishing phrasing, improving clarity and conciseness, and suggesting alternative ways to express our messages.

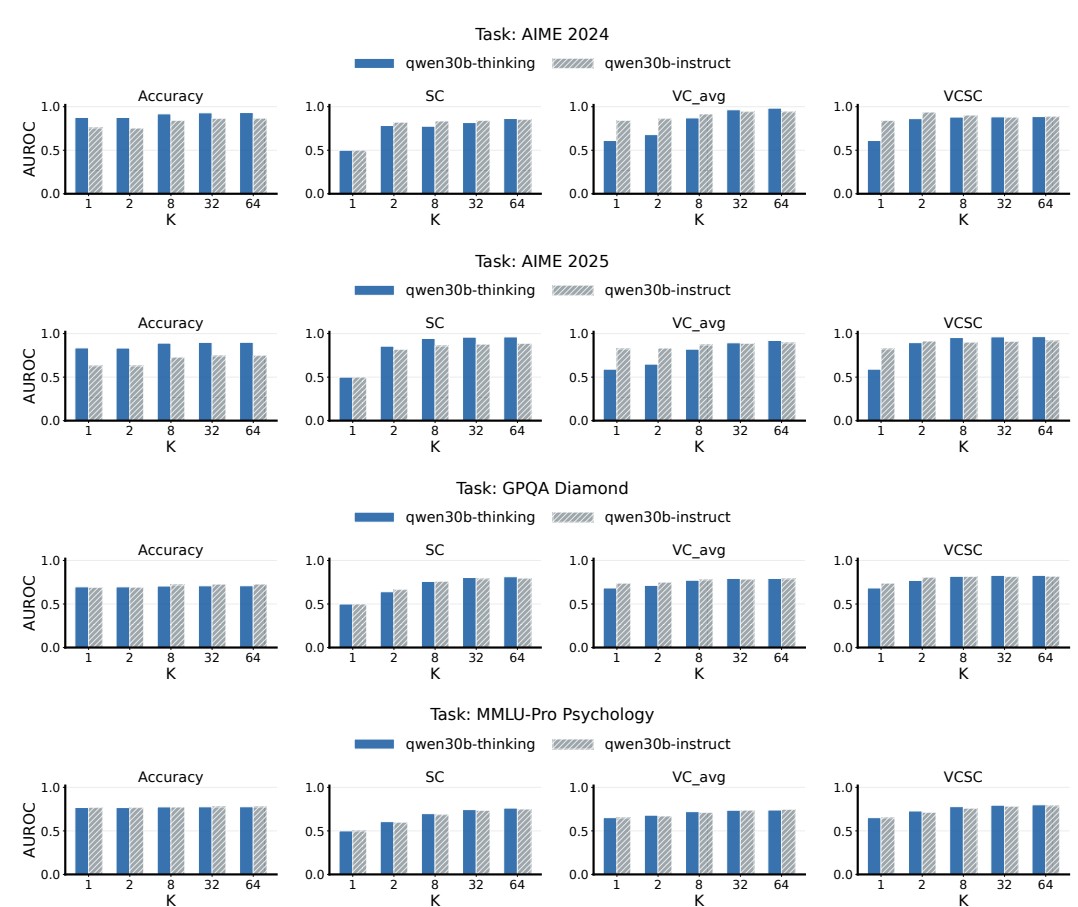

Figure 5: Ablating reasoning mode. Shown separately for each task.

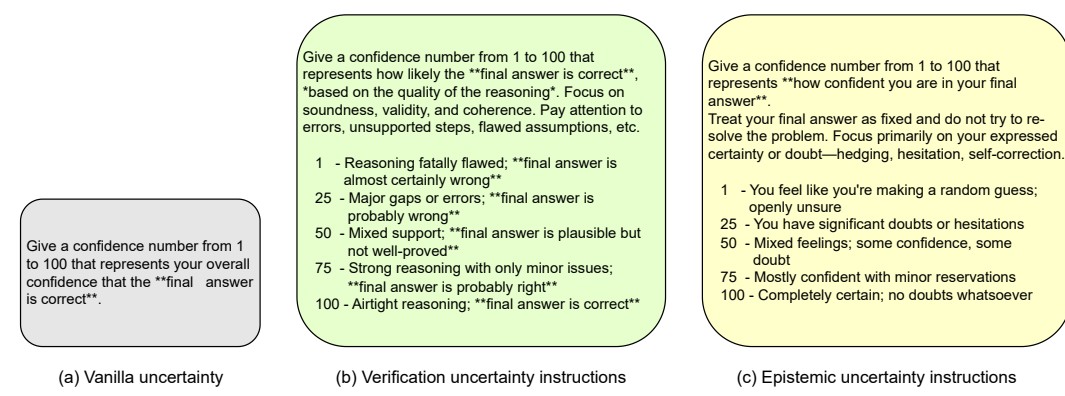

(a) Vanilla uncertainty

(b) Verification uncertainty instructions

(c) Epistemic uncertainty instructions

Figure 6: Overview of uncertainty instructions prompts defining VC methods. (a) Vanilla uncertainty instruction, (b) Verification uncertainty instruction and (c) epistemic uncertainty instruction. Each of the instructions is used both for elicitation and judge methods. For judge method, the epistemic uncertainty instructions are a bit different, as it needs to pay attention to the solver's reasoning trace, not its own.

