# OpenReview forum: "Two Samples Are Enough: Verbal Confidence Meets Self-Consistency in Reasoning LLMs"
_ICLR.cc/2026/Conference — ICLR 2026 Conference Withdrawn Submission_

### Official Review · Reviewer_WKmJ · 2025-10-22

**Soundness:** 3
**Presentation:** 3
**Contribution:** 1
**Rating:** 2
**Confidence:** 4

**Summary:**

This paper evaluates methods for uncertainty quantification with reasoning LLMs, focused on variants of verbalized confidence and self-consistency. From a comparison of three models on 3 math and 2 general reasoning datasets, the authors present a mixed picture, with many methods offering similar AUROC when predicting final answer correctness. The authors additionally evaluate the role of number of sampled traces, but again report a mixed picture where the effect of number of parallel samples varies by model for self-consistency, but appears to positive increase AUROC for average verbalized confidence. Finally, the authors evaluate a combination of self-consistency and averaged verbalized confidence (controlled by a mixing hyperparameter, lambda), and report relatively strong results when using 2 or more sampled traces.

**Strengths:**

* The linear combination of verbalized confidence and self-consistency appears to be relatively insensitive to the mixing hyperparameter lambda, which would support its practical application.
* The choice of AUROC as an evaluation metric is well-motivated, given discriminative power is more important than calibrated estimates in this context.
* The bootstrapping methodology is a welcome addition for improving the estimates of results. However, confidence intervals are surprisingly neglected.

**Weaknesses:**

* The principal contributions of this work are (1) an empirical evaluation of a handful of known methods for uncertainty quantification on select reasoning benchmarks, finding a mixed picture of results, and (2) proposing and evaluating a combination of two existing methods, which demonstrates positive results. I find these contributions to be relatively limited in scope, particularly given the mixed results picture of the empirical evaluation.
* The framing that uncertainty quantification has two main approaches, verbalized confidence and self-consistency, is inaccurate. There are many more families of methods that have been applied to quantifying LLM uncertainty: making use of token output distributions [1], semantic entropy [2], probing approaches [3], etc. If the authors wish to focus on just two approach families, the reason for this reduced focus should be clearly motivated. Equally, the related work section would benefit from significant expansion to cover alternative approaches.
* The model names are needlessly truncated. It would be much clearer to use the full model names throughout, e.g. "GPTOSS-20B (high)", "DeepSeek-R1-0528-Qwen3-8B". Referring to this latter model as "DeepSeek8B" is particularly confusing, given this is really a Qwen3 model with a little reasoning post-training on distilled traces.
* The MMLU Pro evaluation is only on a limited set of domains, with limited justification for why other domains were dropped. Given the highly variable results by MMLU-Pro subject (Table 2), it would be helpful to have a full picture of MMLU Pro performance. Ideally, the overall MMLU-Pro performance would be used for the Overall Avg, as the current approach of averaging a single GPQA-Diamond with several MMLU-Pro topics skews results in favor of MMLU-Pro.
* As mentioned above, given the bootstrapping methodology employed, where are no confidence intervals reported? This would go a long way towards showing whether the differences between methods are meaningful.

### References
1. https://arxiv.org/abs/2508.15260
2. https://arxiv.org/abs/2302.09664
3. https://arxiv.org/abs/2406.15927

**Questions:**

1. What is an "epoch" referring to in this context (line 221)?

---

### Official Review · Reviewer_H24z · 2025-10-29

**Soundness:** 2
**Presentation:** 2
**Contribution:** 2
**Rating:** 4
**Confidence:** 3

**Summary:**

In this work, the authors present a systematic evaluation of different verbalized confidence methods in large reasoning models, and provide analysis based on empirical results. They also propose a hybrid method by combining verbalized confidence and self-consistency, which is shown to be effective in improving AUROC with few samples.

**Strengths:**

1. The authors provide a systematic evaluation of the given methods with extensive empirical results.
2. Based on the experiment results, the authors offer empirical insights and practical guidelines for using the different methods.

**Weaknesses:**

1. One major weakness of this paper is the lack of novelty in its methodology: the methods evaluated in this paper are common established baselines, except for VCSC. The proposed VCSC is a simple linear combination of VC and SC, which is conceptually straightforward and lacks theoretical insights.
2. The experiments focus on three mid-size open-source reasoning models. How will the methods perform when applied to larger frontier or black-box models (e.g., GPT-4, Gemini)?
3. The authors allow the reasoning trace up to 60k tokens for generation and another 60k tokens for judging, which is quite long and expensive in budget. It is unclear whether the long traces are necessary.

**Questions:**

1. Table 9 in the appendix shows a clear formatting problem.
2. What's the motivation to average the confidence scores of different traces in $VC_{avg}$? For example, if some traces have high confidence and others have low confidence, taking an average would result in a moderate-level confidence, which does not represent the actual diverged confidences.

---

### Official Review · Reviewer_gcw3 · 2025-10-29

**Soundness:** 2
**Presentation:** 2
**Contribution:** 1
**Rating:** 2
**Confidence:** 5

**Summary:**

This paper studies verbalized confidence and self-consistency on mathematical, natural, and science tasks. The authors introduce a hybrid method that can improve performance with only two samples.

**Strengths:**

1. The methodology presented in this paper is clear and accessible. The concept of integrating two distinct confidence estimation approaches is straightforward.
2. The proposed method is notably simple, which may facilitate its practical application.

**Weaknesses:**

1. The combined method is not actually novel and lacks motivation. It is too trivial to combine two effective methods to achieve further performance improvements.
2. The effectiveness of the proposed method should be further explored because VCSC does not show significant gains on major datasets, and with increases in K, the performance of VCSC does not always improve. Therefore, it is important to further understand the scenarios in which it works or does not work.
3. The organization of this paper is a bit strange. The authors first discuss all the verbalized confidence in Section 5.1 and then present the experimental results of VCSC in the following sections. It is confusing whether this paper aims to provide a systematic evaluation or to introduce a novel method. It seems that the workload for both sides is not sufficient.
4. The results of ECE and NLL can only be found in the Appendix, while they are introduced with extensive text in the main paper. It is confusing why these metrics are heavily discussed without reporting corresponding results in the main paper.
5. Important related works are missing. Generally, this paper discusses how to combine SC and other confidence estimations. As detailed in their method, it does not really matter if the verbalized confidence is replaced with other informative signals. [1] theoretically and empirically studies how to combine self-consistency, perplexity, and rewards together. [2] presents empirical results showing that other confidence measures can help SC. The authors should carefully discuss or compare their work with these related studies.

**Reference**

[1] A Theoretical Study on Bridging Internal Probability and Self-Consistency for LLM Reasoning. NeurIPS 2025.

[2] Confidence Improves Self-Consistency in LLMs. ACL (Findings) 2025.

**Questions:**

Please refer to the `weaknesses` section. Moreover,

1. Are VC methods strongly related to the capacity of the base model? How does the performance change when the base model does not perform well on the target dataset?
2. Why can SC and VC be directly combined to obtain a novel metric? Do they face the challenge of magnitude calibration?
3. If two answers have the same confidence from VC methods, how are they ranked when computing AUROC, given that there are few options for selecting confidence?
4. How does the sampling temperature affect the results of VC and VCSC methods?

---

### Official Review · Reviewer_d9RG · 2025-10-30

**Soundness:** 2
**Presentation:** 3
**Contribution:** 2
**Rating:** 4
**Confidence:** 3

**Summary:**

This paper presents the first systematic study of uncertainty estimation in Large Reasoning Models (LRMs), comparing six Verbalized Confidence (VC) methods, Self-Consistency (SC), and their hybrid combination, VCSC. The experiments reveal that while advanced VC prompting strategies bring little or inconsistent improvement, the simple VCSC hybrid delivers substantial benefits, boosting AUROC by over 10 points with only two samples. These findings highlight that even minimal sampling can significantly enhance the reliability of model confidence, offering a practical and effective recipe for safer deployment of reasoning LLMs.

**Strengths:**

- The paper is clearly written and well-structured, making it easy to follow.
- The experimental setup and evaluation protocol are carefully designed and transparently explained.
- Extensive experiments across multiple benchmarks and reasoning models provide strong empirical evidence for the effectiveness of the proposed VCSC hybrid.
- The study offers practical insights and clear guidelines for improving uncertainty estimation in reasoning LLMs.

**Weaknesses:**

- The technical novelty is relatively limited, as the proposed VCSC method is a straightforward combination of existing VC and SC techniques.
- The source of the large AUROC improvement at small sample sizes (e.g., *K* = 2) is not clearly explained. The paper should explicitly clarify why such a substantial gain is achieved without a corresponding improvement in accuracy.
- Since $\text{VC}_{\text{avg}}$ approaches VCSC performance at larger *K*, the paper should include a cost–benefit analysis to highlight the practical advantage of VCSC under realistic computational budgets.

**Questions:**

1. The paper emphasizes the strong performance of VCSC with small *K*, but more evidence should be provided to demonstrate its practical importance, for example, through comparisons of inference time or computational cost.
2. In Table 3, $\text{VC}_{\text{avg}}$ outperforms VCSC when *K* = 32 and 64 on *qwen30b (AIME 2024)* and *deepseek8b (AIME 2025)*. Could the authors clarify why this reversal occurs and whether it indicates diminishing complementarity between VC and SC at larger sampling budgets?
3. Since Expected Calibration Error (ECE) and Negative Log-Likelihood (NLL) are more standard metrics for uncertainty estimation than AUROC [1], it would strengthen the paper to include these baseline results for completeness and fair comparison.

[1] Tian et al., ``Just Ask for Calibration: Strategies for Eliciting Calibrated Confidence Scores from Language Models Fine-Tuned with Human Feedback.'' (2023)

---

### Note · Authors · 2025-12-04

**Comment:**

We thank the reviewers and the area chair for their time and feedback. Following internal discussion, we are withdrawing the submission with a clear vision for its revision.

**Withdrawal Confirmation:**

I have read and agree with the venue's withdrawal policy on behalf of myself and my co-authors.